# Tissue engineered vascular grafts are resistant to the formation of dystrophic calcification

Mackenzie E. Turner [1,2,17], Kevin M. Blum[1,3,17], Tatsuya Watanabe [1,17], Erica L. Schwarz[4,17], Mahboubeh Nabavinia [1,17], Joseph T. Leland [1], Delaney J. Villarreal [1,3,5], William E. Schwartzman [1,3], Ting-Heng Chou [1,6], Peter B. Baker[7], Goki Matsumura[8], Rajesh Krishnamurthy [9], Andrew R. Yates [10,11], Kan N. Hor[10,11], Jay D. Humphrey [4], Alison L. Marsden[12], Mitchel R. Stacy[1,13,14], Toshiharu Shinoka [1,11,15] & Christopher K. Breuer [1,13,16] ✉

Advancements in congenital heart surgery have heightened the importance of durable biomaterials for adult survivors. Dystrophic calcification poses a significant risk to the long-term viability of prosthetic biomaterials in these procedures. Herein, we describe the natural history of calcification in the most frequently used vascular conduits, expanded polytetrafluoroethylene grafts. Through a retrospective clinical study and an ovine model, we compare the degree of calcification between tissue-engineered vascular grafts and polytetrafluoroethylene grafts. Results indicate superior durability in tissue-engineered vascular grafts, displaying reduced late-term calcification in both clinical studies ($p < 0.001$) and animal models ($p < 0.0001$). Further assessments of graft compliance reveal that tissue-engineered vascular grafts maintain greater compliance ($p < 0.0001$) and distensibility ($p < 0.001$) than polytetrafluoroethylene grafts. These properties improve graft hemodynamic performance, as validated through computational fluid dynamics simulations. We demonstrate the promise of tissue engineered vascular grafts, remaining compliant and distensible while resisting long-term calcification, to enhance the long-term success of congenital heart surgeries.

All currently available biomaterials used in cardiovascular surgery are susceptible to dystrophic calcification[1], which arises from pathological biomineralization of prosthetic materials when used as vascular grafts, cardiovascular patches, or replacement heart valves[2]. Dystrophic calcification has been linked to the development of vascular graft failure[3]; it stiffens the prosthetic material, increasing the compliance mismatch between the graft and native blood vessel which, in turn, compromises hemodynamic performance. Furthermore, dystrophic calcification can progress and impinge on the lumen, contributing to the formation of critical stenosis that requires treatment with angioplasty, stenting, or surgical graft replacement[3].

Calcification may also complicate any necessary re-operation through calcific attachment to surrounding anatomy. The incidence, time of onset, and severity of dystrophic calcification vary based on the chemical and physical properties of the biomaterial and its clinical application[1]. In addition, the recipient's age and underlying disease state can also impact the development and severity of dystrophic calcification[2]. Such calcification is particularly problematic in pediatric congenital heart surgery where it represents the leading cause of late-term biomaterial failure[1]. Avoidance of prosthetic biomaterials by using autologous tissue prevents biomaterial-related complications and is currently viewed as the gold standard.

Unfortunately, adequate autologous tissue is often not available for surgical reconstruction.

Tissue engineering provides a method for creating additional autologous neotissue for repairing or replacing tissues that are diseased, damaged, or congenitally absent[4]. The overriding premise of our work is that autologous neotissue will perform similarly to native tissue and better than currently available biomaterials. We developed a tissue-engineered vascular graft (TEVG) for use in children with single ventricle cardiac anomalies undergoing extracardiac modified Fontan surgery, in which a vascular conduit is used to connect the inferior vena cava (IVC) to the pulmonary artery. Reports of this first-in-human clinical trial can be found elsewhere[5–7].

Herein, we compare the late-term formation of dystrophic calcification in these TEVGs to the most commonly used conduit in the Fontan operation, poly(tetrafluoroethylene), or PTFE, grafts[8]. First, we define the natural history of dystrophic calcification in the PTFE graft by analyzing pathological specimens from a single institutional cohort of Fontan patients. Next, we retrospectively evaluate and compare the formation of dystrophic calcification between TEVGs and PTFE grafts in a matched cohort clinical study. We then validate our findings by quantifying and comparing dystrophic calcification and biomechanical properties of TEVGs versus PTFE grafts in a clinically relevant large animal model of the Fontan operation[9–11]. Finally, we evaluate the impact of compliance mismatch on graft hemodynamics. Despite similar geometries under normal flow conditions, the difference in compliance between the TEVG and PTFE grafts resulted in significantly different geometries during imposed bolus flow conditions. We use computational fluid dynamics to discover differential hemodynamic performance based on geometrically different 3D computational models of TEVG and PTFE graft-host IVC segments under steady flow rates corresponding to normal and bolus flow.

## Results

### Describing the natural history of calcification in PTFE grafts

We identified all gross pathological specimens available from the Pathology Department at Nationwide Children's Hospital in Columbus, Ohio for children who had undergone a primary extracardiac modified Fontan operation using a PTFE conduit. In total, there were 5 specimens ranging from less than 1 day to more than 20 years after implantation (Table 1). All 4 of the conduits that had been implanted for more than 1 day were grossly calcified upon visual inspection. We next obtained computed tomography (CT) scans of the pathological specimens as well as a phantom of a PTFE graft (prior to implantation) created by embedding the graft in 1% agar. An established CT image threshold of ≥130 Hounsfield units (HUs) was used to identify and non-invasively quantify relative calcium mass for each specimen[12]. The relative calcium mass of the PTFE graft phantom (35 HU*cm$^3$) and the PTFE specimen that had been implanted for <1 day (21 HU*cm$^3$) were both low, whereas the values from the 4 specimens that had been implanted for more than 1 day measured between 97 HU*cm$^3$ and 910 HU*cm$^3$, demonstrating

evidence of significant calcium burden[12]. We also performed histological examinations of samples from each PTFE graft; Von Kossa staining confirmed the presence of diffuse calcification in all 4 specimens implanted for more than 1 day. Comparison of the relative calcium mass value to the histology-derived calcium area (by Von Kossa staining) using simple linear regression revealed a strong correlation ($R^2 = 0.9016$, $p = 0.0135$), confirming that relative calcium mass is a valid metric for detecting dystrophic calcification in PTFE grafts. A comparison of the degree of calcification versus time of implantation revealed that dystrophic calcification forms within months of implantation and tends to progress over time (Fig. 1) (Table 1).

### Comparing calcium burden of TEVGs and PTFE Fontan conduits in humans

Next, we performed a retrospective clinical study comparing in vivo CT data obtained from TEVG patients from our original clinical trial[5–7] to a matched cohort of patients receiving PTFE grafts. We studied 3 late-term CT scans from TEVGs imaged between 7.4 and 13.2 years after implantation and compared results to an age-matched cohort of 6 patients who received PTFE grafts and were imaged between 2 months and 16.2 years after implantation [TEVG mean implantation at 5.3 years (range 2.1–11.1 years); PTFE mean implantation of 6.2 years (range 3.3–14.3 years)]. There was no statistically significant difference between the ages of both groups ($p = 0.357$, Mann Whitney U test). We measured and compared the relative calcium mass between both cohorts which showed an average value of 11 HU*cm$^3$ in TEVGs versus 690 HU*cm$^3$ in PTFE grafts ($p < 0.001$), thus demonstrating that TEVGs resist formation of dystrophic calcification over many years while PTFE grafts are uniformly susceptible to severe dystrophic calcification when used as extracardiac Fontan conduits even at durations <1 year (Fig. 2) (Table 2).

### Confirming the retrospective clinical findings using an ovine model

To validate our retrospective clinical findings, we performed a prospective head-to-head comparison of the formation of dystrophic calcification in PTFE grafts ($n = 9$) versus TEVGs ($n = 11$) implanted as intrathoracic IVC interposition grafts in sheep[9–11]. The grafts were implanted in juvenile sheep between 4–8 months of age. We evaluated for dystrophic calcification of the grafts using non-contrast CT imaging between 4.6 and 5.7 years after implantation (the typical life span of sheep is approximately 10 years). The average calcium mass in the TEVG group was 0.1 HU*cm$^3$ (range 0.0 HU*cm$^3$–0.4 HU*cm$^3$), suggesting these TEVGs were free from long-term dystrophic calcification. By contrast, the average calcium mass value in the PTFE graft group was 269 HU*cm$^3$ (range 65 HU*cm$^3$–496 HU*cm$^3$), confirming that PTFE grafts are susceptible to severe late-term dystrophic calcification (Fig. 3) (Table 3). The difference between values for the TEVGs and PTFE grafts was highly significant ($p < 0.0001$).

### Comparing in vivo compliance and distensibility of TEVGs and PTFE grafts in sheep

We performed a fluid volume challenge study to measure the compliance of TEVGs and PTFE grafts in vivo in the ovine intrathoracic IVC interposition graft model. We calculated in vivo compliance and distensibility using dimensions from 3D rotational angiograms and catheter-based pressure measurements for conduits before and after administration of a bolus of 20 ml/kg of isotonic fluid. Administration of the bolus increased the mean pressure within the conduit by (11.40 ± 3.15 mmHg), regardless of the type of the graft. Our analysis demonstrated that the TEVGs were significantly more compliant (0.06 ± 0.03 cm$^2$/mmHg versus 0.002 ± 0.002 cm$^2$/mmHg, $p = 0.0002$) and distensible (146 ± 95%/100 mmHg vs 7.8 ± 10.65/100 mmHg, $p < 0.0001$) when compared to the PTFE grafts. (Fig. 4). We correlated the compliance and distensibility to the calcium mass which revealed a nonlinear

**Table 1 | Assessment of calcium burden in pathological specimens (relative calcium mass) and histological sections (% stained) for human PTFE graft**

| Human pathology specimen | Graft Type | Age at implantation (Years) | Duration of graft implantation prior to CT (Years) | Relative calcium mass (HU*cm$^3$) | % area positive stain |
|---|---|---|---|---|---|
| 1 | PTFE | 4.0 | 0.0 (<1 day) | 21 | 0 |
| 2 | PTFE | 2.8 | 0.3 | 174 | 5.2 |
| 3 | PTFE | 2.3 | 0.3 | 97 | 0.9 |
| 4 | PTFE | 3.0 | 20.0 | 910 | 10.9 |
| 5 | PTFE | 1.7 | 20.6 | 571 | 11.7 |

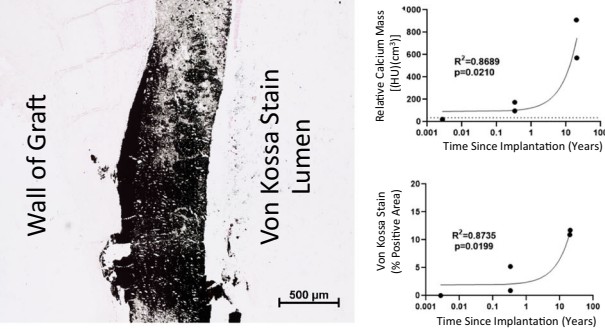

**Fig. 1 | PTFE grafts form dystrophic calcification when used as extracardiac conduits in modified Fontan surgery.** Representative (**A**) gross, (**B**) CT, and (**C**) histological images of late-term extracardiac Fontan PTFE grafts demonstrate marked calcification (*n* = 2, images correspond to Pathology Specimen 4 in Table 1). Yellow dotted outline indicates area of graft (note: calcium is radiopaque and appears white on CT scan). **D** Quantified CT-detectable calcification (horizontal dotted line represents the relative calcium mass of a PTFE phantom) as a function of time following implantation (top; $R^2$ = 0.8689, *p* = 0.0210) and histological detection of calcification (% area positive (black=calcified) on Von Kossa stain) since implantation (Simple Linear Regression $R^2$ = 0.8735, *p* = 0.0199) demonstrate both the early onset (within months) and late rapid progression of dystrophic calcification. See Table 1 or the source data file for specific values.

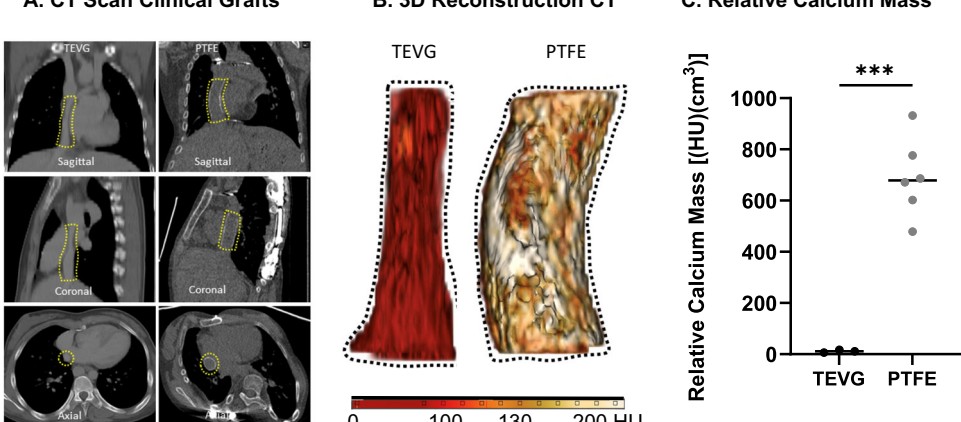

**Fig. 2 | Evaluation and comparison of dystrophic calcification in PTFE grafts versus TEVGs in a matched clinical cohort.** Representative (**A**) CT images of a TEVG and a PTFE graft implanted in human subjects as extracardiac Fontan conduits. Yellow dotted outline indicates the region of interest. **B** 3D reconstructions of the TEVG and PTFE grafts showing diffuse increase in radiodensity in PTFE graft compared to TEVG. **C** Quantification of CT-derived relative calcium mass values in human subjects demonstrated that PTFE grafts (*n* = 6) were significantly more calcified than TEVGs (*n* = 3), as determined using an unpaired two-sided *t* test with Welch's correction (*p* < 0.001). See Table 2 for specific values. The values are also included in the source data file.

correlation (compliance versus calcium mass ($R^2$ = 0.5827) and distensibility versus calcium mass ($R^2$ = 0.4887)) (Supplementary Fig. 1)

## Determining the impact of compliance mismatch on hemodynamic performance

To better understand the implications of reduced compliance matching on conduit performance, we developed idealized 3D geometric models of the PTFE grafts and TEVGs plus host IVC and performed computational fluid dynamics simulations. We focused on multiple hemodynamic metrics, including flow fields, pressure drop across the graft, flow-induced wall shear stress (WSS), and power loss, which was calculated as

$$\text{Power loss} = 1 - \frac{\int_{A_{out}} \left(p + \frac{1}{2}\rho u^2\right) \boldsymbol{u} \cdot \mathrm{d}A}{\int_{A_{in}} \left(p + \frac{1}{2}\rho u^2\right) \boldsymbol{u} \cdot \mathrm{d}A} \tag{1}$$

where $A_{in}$ is the inlet the control surface area, $A_{out}$ is the outlet control surface area, $u$ is the magnitude of the velocity, $\boldsymbol{u}$ is the velocity in the normal direction to the control surface, $p$ is the static pressure, and $\rho$ is the fluid density.

These models were informed by hemodynamic and compliance data from our ovine studies. Under baseline conditions, both the initially diameter-matching PTFE graft and TEVG produced a pressure drop of 0.22 mmHg and a power loss (Eq. 1) of 3.4% with WSS constant throughout the graft region at 9.2 dynes/cm². Under bolus flow conditions, both the IVC and TEVG increased in diameter to accommodate the increase in flow, and the pressure drop decreased to 0.15 mmHg thus reducing power loss by 62% (to 1.3%) while WSS remained constant throughout the graft complex at 8.02 dynes/cm². Under bolus conditions, distension of the IVC but not PTFE graft created a "transient stenosis" that increased the pressure drop to 2.76 mmHg and power loss by 8.9% to 3.7%; the WSS thus varied substantially across the graft complex with a maximum WSS of 196.14 dynes/cm², which occurred at the proximal graft anastomosis, and a minimum WSS of 0.02 dynes/cm², which occurred just outside of the graft anastomoses in the IVC. In addition, the lack of distension of the PTFE graft associated with the compliance mismatch resulted in a downstream region of recirculation and flow reversal that was not present at pre-bolus baseline conditions (Fig. 5).

## Discussion

Herein, we quantify the susceptibility of PTFE grafts to the formation of progressive dystrophic calcification when used as extracardiac Fontan conduits. We further show that the TEVGs resisted the formation of dystrophic calcification over many years in both a retrospective clinical study and a clinically relevant large animal study. Using the large animal model, we also demonstrate better compliance matching and distensibility of the TEVGs compared to the PTFE grafts. Finally, computational simulations of hemodynamics within idealized models of a representative TEVG and PTFE graft, informed with data from our animal studies, showed how improved compliance matching can result in superior graft performance by preventing the formation of a relative stenosis during periods of increased flow. In the setting of compliance mismatch, this relative stenosis worsens the pressure drop and power loss across the graft in addition to altering wall shear stress and causing areas of recirculation and reversal of flow. The clinical implications of these results are manifold.

While dystrophic calcification affects all fields of cardiovascular surgery, its impact on the field of congenital heart surgery is particularly problematic[1]. As survival rates following congenital heart surgery continue to improve, the number of long-term survivors continues to grow[13]. The number of adult survivors now exceeds the number of pediatric patients with critical cardiac anomalies[14]. Thus, dystrophic calcification represents a rapidly emerging medical problem and there is a great need to develop biomaterials with improved durability.

The management of patients born with single ventricle disease illustrates this problem. As early surgical survival continues to improve, the focus has shifted from reducing mortality to improving quality of life. Fontan patients can now expect to live well into their third decade of life[15]. Despite this improved survival, there is a paucity of literature describing graft performance beyond 10 years after implantation[8]. Currently, available prosthetic vascular grafts for use in the Fontan operation include both biological and synthetic conduits. Synthetic grafts are used more commonly than biological conduits due to the high incidence of dystrophic calcification in current biological vascular conduits, often within the first few years after implantation[8]. Synthetic options include PTFE grafts and poly(ethylene terephthalate), or PETE, grafts. PTFE grafts have supplanted the use of PETE grafts and represent the current standard of care largely due to reports of a high incidence of stenosis in PETE grafts due to neointimal hyperplasia[16,17]. Nevertheless, PTFE grafts are still associated with a risk of graft-related complications. One study reported that extracardiac PTFE Fontan conduits are at particularly high risk for dystrophic calcification, with 100% of grafts (7/7) affected within just $1.6 \pm 0.4$ years of implantation[12]. Indeed, there is a growing body of literature demonstrating dystrophic calcification as a leading cause of late-term PTFE graft failure requiring reintervention[8,18]. Dystrophic calcification is not only the underlying cause of most cases of late-term stenosis, it also increases complications associated with treating stenosis whether using interventional angioplasty, stenting, or open surgical graft replacement[18]. Furthermore, as demonstrated herein, dystrophic calcification exacerbates the compliance mismatch between the conduit and the native vessel, which worsens hemodynamic performance by increasing power loss and altering wall shear stress. Decreased wall shear stress is associated with increased the risk of thromboembolic complications, the most common serious adverse events associated with Fontan surgery. As this patient population continues to age, there is a pressing need to monitor graft performance over the lifetime of all patients and to develop biomaterials with improved durability.

Calcification is a relatively understudied complication associated with Fontan conduits and the progression to elevated conduit pressure. Calcification can also progress and cause stenosis, but this is a less frequent occurrence[3]. Although none of the grafts in this study demonstrated clinical failure, calcification was present and our findings suggest that decreased compliance is detrimental to long-term

**Table 2 | Assessment of calcium burden comparing TEVG (left) to PTFE (right) grafts using CT imaging in a matched clinical cohort study**

| Graft type | Age at implantation (years) | Duration of implantation prior to CT (years) | Relative calcium mass (HU*cm³) |
|---|---|---|---|
| *Human PTFE Subjects* | | | |
| CT PTFE1 | 3.4 | 0.2 | 478 |
| CT PTFE2 | 6.5 | 2.2 | 602 |
| CT PTFE3 | 14.3 | 6.6 | 932 |
| CT PTFE4 | 6.1 | 8.1 | 671 |
| CT PTFE5 | 3.4 | 11.5 | 680 |
| CT PTFE6 | 3.3 | 16.2 | 776 |
| *Mean* | 6.2 | 7.5 | 690 |
| *Human TEVG Subjects* | | | |
| CT TEVG1 | 11.1 | 7.4 | 11 |
| CT TEVG2 | 2.1 | 13.2 | 6 |
| CT TEVG3 | 2.7 | 13 | 17 |
| *Mean* | 5.3 | 11.2 | 11 |

### A. CT Scan Ovine Grafts

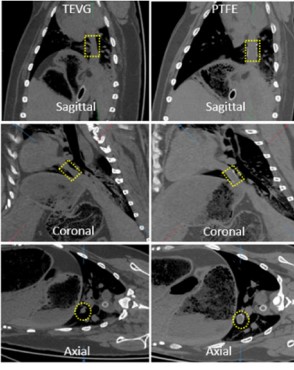

### B. 3D Reconstruction CT

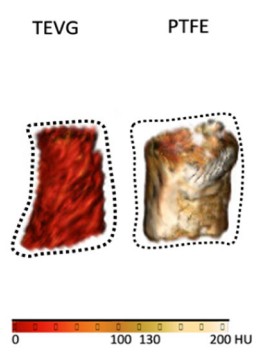

### C. Relative Calcium Mass

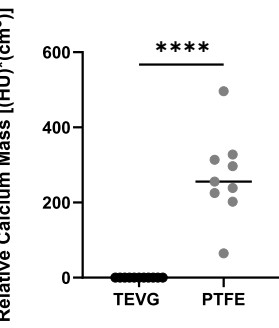

**Fig. 3 | Evaluation and comparison of dystrophic calcification in PTFE versus TEVG 4–6 years after implantation as IVC-interposition grafts in an ovine model.** Representative (**A**) non-contrast CT images of TEVG and PTFE graft (region of interest outlined with dotted yellow line). **B** 3D reconstructions of TEVG and PTFE graft (conduit outlined in dotted black line) (presence of calcification determined using ≥130 HU threshold). **C** Quantification of the CT-derived relative calcium mass in the ovine model demonstrated that PTFE grafts ($n = 9$) were significantly more calcified than TEVGs ($n = 11$), as determined using a Mann–Whitney $U$ test ($p < 0.0001$). See Table 3 for specific values. Source data are provided as a source data file.

**Table 3 | Assessment of the burden of calcification comparing TEVGs to PTFE grafts using CT imaging in an ovine model**

| Graft type | Age at implantation (years) | Duration of implantation prior to CT (years) | Relative calcium mass (HU*cm³) |
|---|---|---|---|
| *Ovine PTFE subjects* | | | |
| PTFE1 | 0.6 | 4.6 | 328 |
| PTFE2 | 0.7 | 4.6 | 202 |
| PTFE3 | 0.5 | 4.6 | 65 |
| PTFE4 | 0.7 | 4.6 | 297 |
| PTFE5 | 0.5 | 4.6 | 225 |
| PTFE6 | 0.5 | 4.6 | 496 |
| PTFE7 | 0.6 | 4.7 | 314 |
| PTFE8 | 0.6 | 4.7 | 239 |
| PTFE9 | 0.5 | 4.7 | 256 |
| *Mean* | 0.6 | 4.6 | 269 |
| *Ovine TEVG subjects* | | | |
| TEVG1 | 0.4 | 4.8 | 0.4 |
| TEVG2 | 0.4 | 4.8 | 0.0 |
| TEVG3 | 0.4 | 4.8 | 0.0 |
| TEVG4 | 0.3 | 4.8 | 0.0 |
| TEVG5 | 0.5 | 4.8 | 0.1 |
| TEVG6 | 0.3 | 5.2 | 0.0 |
| TEVG7 | 0.3 | 5.2 | 0.1 |
| TEVG8 | 0.2 | 5.2 | 0.0 |
| TEVG9 | 0.3 | 5.2 | 0.1 |
| TEVG10 | 0.3 | 5.3 | 0.0 |
| TEVG11 | 0.3 | 5.7 | 0.3 |
| *Mean* | 0.3 | 5.1 | 0.1 |

performance. The mechanisms underlying the formation of dystrophic calcification on or within biomaterials implanted in the cardiovascular system remain incompletely understood. Once considered to be a passive degeneration, dystrophic calcification is now seen as a complex process actively regulated by several factors including material composition, mechanical stress, and immune response. Extracellular matrix damage, subclinical thrombosis, metabolic disorders, and oxidative stress have also been linked to dystrophic calcification[2,19]. Many of the signaling pathways involved in this process are the same as those involved with the physiological calcification of bone, suggesting that the mesenchymal and inflammatory cells that normally maintain a balance between pro-calcific and anti-calcific states are out of balance[2,20,21]. The time of onset and severity of dystrophic calcification vary with the chemical and physical properties of the biomaterial[19]. Biological materials, including processed or unprocessed homografts or xenografts, are often more susceptible to dystrophic calcification compared to materials such as metals or synthetic polymer[22]. In addition, the site of implantation dramatically affects the formation of dystrophic calcification. Replacement heart valves elicit a greater response than vascular conduits or patches, while biomaterials implanted in lower-pressure circulations (pulmonary, Fontan, or venous systems) induce earlier onset and greater dystrophic calcification than those implanted in the arterial circulation. Finally, the graft recipient (host) also impacts dystrophic calcification. Younger patients are more susceptible to dystrophic calcification while diverse underlying disease or genetic defects predispose individuals to developing dystrophic calcification[22,23]. A variety of strategies have been developed to reduce or slow the formation of dystrophic calcification with varying degrees of success. Nevertheless, all biomaterials are susceptible to dystrophic calcification and while some reductions have been realized, there are still no optimal biomaterials for use in congenital heart surgery[1].

This study was motivated by the observation that autologous tissue is inherently more resistant to dystrophic calcification than non-autologous tissue and current biomaterials[1]. We previously demonstrated that our poly(glycolic acid)-based TEVGs can transform from a biodegradable scaffold seeded with bone-marrow-derived mononuclear cells into an autologous neovessel with a functional intima and media and growth capacity[11]. Fabricated from a knitted poly(glycolic acid) fiber tube and coated with a 50:50 copolymer sealant of poly(-caprolactone) and poly(lactic acid), this TEVG is designed to degrade by hydrolysis, losing its biomechanical integrity approximately 2 months after implantation while total fiber degradation takes approximately 6 months[10]. This TEVG scaffold induces a foreign body response upon implantation. The infiltrating monocytes and macrophages induce the ingrowth of endothelial cells and smooth muscle cells from the neighboring vascular wall along the luminal surface of the scaffold via paracrine signaling[24,25]. Absence of inflammation blocks neotissue formation while excessive inflammation leads to the formation of stenosis[26]. Interestingly the seeded cells disappear shortly after implantation and do not contribute directly to the vascular neotissue; they instead serve an immunomodulatory role[25]. Ultimately, an inflammation-driven, mechano-mediated process[27] gives rise to an autologous neovessel without any synthetic components by about 6–12 months after implantation[11]. Structurally and functionally similar to the native blood vessel into which it is implanted[11], we demonstrated herein that this TEVG also resists the formation of dystrophic calcification similar to native tissue. Importantly, this study focuses on Fontan conduits, a low-pressure high-flow system. These findings may not be directly applicable to high-pressure systems such as arterial circulation.

There are several limitations to this study. Due to the retrospective nature of the clinical study, and the lack of routine in vivo assessment or examination of explanted grafts in practice, the number of specimens available for analysis was limited. We were limited by the samples available to us, which raises the possibility of selection bias. However, we did not exclude any available samples on the basis of age, sex, race, etc. Further prospective study is needed in clinical populations to substantiate our findings. Non-contrast CT scans represent the most accurate method for detecting dystrophic calcification[12], but inherent risks of radiation exposure in children typically limits the use of CT. Additionally, we had few clinical CT scans for comparison due to the routine use of intravascular contrast in most of clinical CT scans in patients with PTFE grafts. The use of contrast resulted in our inability to analyze calcium burden due to the presence of a high concentration of intraluminal contrast in the Fontan conduit at the time of image acquisition, which rendered conventional CT image thresholding techniques for detecting and quantifying calcium impossible. We used a smaller number of contrast-enhanced studies that did not have contrast in the Fontan circuit due to the phase of contrast enhancement and performed established segmentation and thresholding to generate CT attenuation values of the conduit wall. We do not feel that the results were influenced by this approach because the presence of intraluminal contrast has been shown to underestimate the degree of dystrophic calcification when using conventional CT thresholding techniques[12], thus likely underestimating the severity of PTFE graft calcification. We augmented the clinical study with a carefully performed animal study using the intrathoracic IVC interposition model. We developed this model as a surrogate for the Fontan model due to high mortality rates associated with performing a Fontan operation on an animal with a structurally normal heart[9]. While we have used data generated using this model to support our clinical studies, differences in hemodynamics between the venous circulation and Fontan circulation may impact the formation of dystrophic calcification. Lastly, our preliminary computational studies of the hemodynamics used highly idealized geometric models. Development of patient-specific models

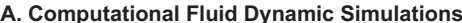

**A. Angiography**

**B. Midgraft Compliance**

**C. Midgraft Distensibility**

**Fig. 4 | Evaluation and comparison of in vivo compliance and distensibility in PTFE versus TEVG late term after implantation as IVC-interposition grafts in an ovine model. A** Representative images of a 3D rotational angiogram comparing the area of a TEVG versus a PTFE graft before and after a bolus of isotonic fluid. **B** Compliance measurements of the midgraft comparing TEVGs ($n = 11$) to PTFE grafts ($n = 9$) (unpaired two-sided $t$ test with Welch's correction $p = 0.0002$).

**C** Comparison of the distensibility of TEVGs ($n = 11$) versus PTFE grafts ($n = 9$) (Mann–Whitney U $p < 0.0001$). Implantation periods were similar and are provided in Table 3. The sheep-to-sheep variability in compliance and distensibility of the TEVGs results from both intrinsic differences in rates of maturation of neotissue in vivo and differences in pressure elevation ($11.4 \pm 3.15$ mmHg) associated with the imposed bolus injections. Source data are provided as a source data file.

**A. Computational Fluid Dynamic Simulations**

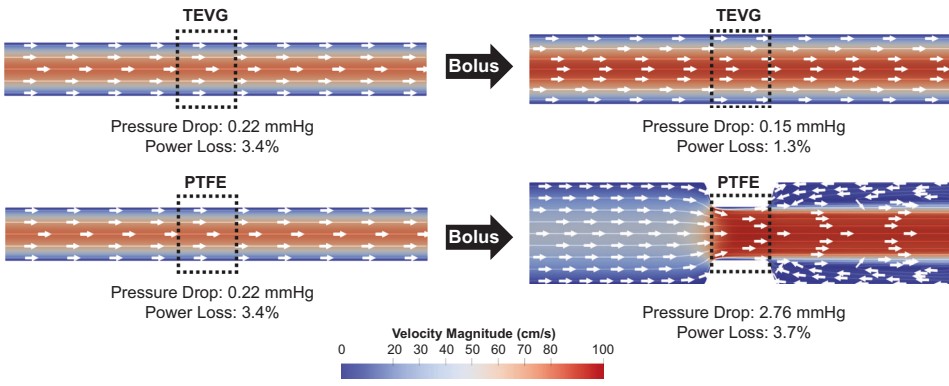

**B. Simulated Wall Shear Stress Along Idealized Grafts**

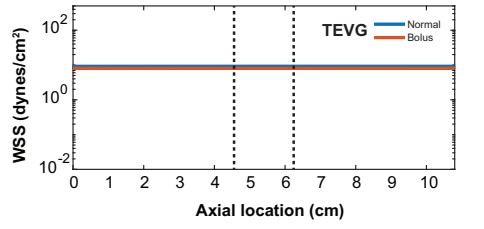

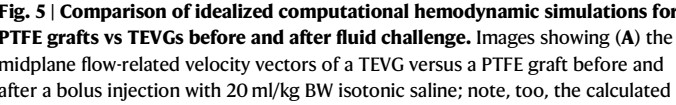

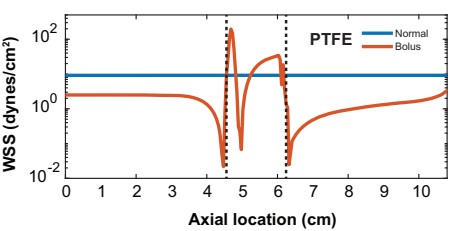

**Fig. 5 | Comparison of idealized computational hemodynamic simulations for PTFE grafts vs TEVGs before and after fluid challenge.** Images showing (**A**) the midplane flow-related velocity vectors of a TEVG versus a PTFE graft before and after a bolus injection with 20 ml/kg BW isotonic saline; note, too, the calculated

pressure drop and power loss across the graft. **B** Wall shear stress (WSS) along the wall of the IVC and interposition graft comparing a representative TEVG to a PTFE graft before and after a fluid challenge. The dotted central domain indicates the graft. Source data for panel B is provided in the source data file.

will be needed to determine the actual impact of dystrophic calcification on pressure, energy loss, and compliance mismatch.

We developed the TEVG in an attempt to create an improved biomaterial for use in congenital heart surgery[10]. We selected the

extracardiac Fontan conduit as our initial clinical target due to its relatively safe risk profile compared to other clinical targets because it requires a large diameter conduit for use in a high-flow, low-pressure circulation. In this setting, the risks of acute graft failure due to

thrombosis or aneurysmal dilation and rupture are minimized. Additionally, the use of TEVGs in a pediatric population takes advantage of the graft's growth potential, thus avoiding the major concern of somatic overgrowth (outgrowing a prosthetic device) in all congenital heart operations[11]. The development of autologous biomaterials that are resistant to dystrophic calcification holds promise for improving graft durability and late-term outcomes for children born with complex congenital cardiac anomalies. Furthermore, living vascular conduits having the ability to repair, remodel, and grow may enable congenital heart surgeons to operate earlier, limiting exposure to chronic hypoxia and volume overload which are known to negatively impact development and impair heart function. We view the present TEVG as an archetype for any tissue-engineered construct (including vascular patches, conduits, or replacement heart valves) designed for use in congenital heart surgery. We contend that the lessons learned from developing and translating this technology will facilitate the development and translation of other tissue-engineered prosthetics.

## Methods

### Ethical statement

All animal studies were performed according to ARRIVE Guidelines and under the approval and guidance of the NCH Animal Welfare and Resource Committee (Approval AR13-00079). Sheep were housed in facilities that are USDA licensed and AAALAC accredited. Housing space for sheep was in accordance with the 2011 Guide for the Care and Use of Laboratory Animals (including HVAC parameters, lighting, and airflow). No animals were euthanized for the purpose of this study.

### Pathological specimens

The specimens were collected at autopsy and consent was obtained to authorize the use of specimens for research purposes. We obtained all de-identified pathological specimens with an extracardiac Fontan PTFE conduit (GORE-TEX®) available within our institution (Nationwide Children's Hospital; IRB STUDY00003038). These samples were not separately consented for this particular research. The majority of them were collected at the time of autopsy – not surgery. They were collected for non-human subjects research under a general autopsy consent, indicating the following: "I authorize the removal, examination, and retention of specimens including tissue, organs (including the brain), fluids, and devices ("Autopsy specimens") for diagnostic, education, quality improvement, and or research purposes". Patient demographics were obtained, gross pathological images were taken, and computed tomography (CT) scans of the entire pathological specimen were acquired with a slice thickness of 0.625 mm, at 450 mA, and 120 kVp using a commercially available clinical scanner (Discovery PET/CT 690, GE Healthcare). After scanning the pathological specimen, a section of the conduit was obtained and Von Kossa staining was carried out. Images were obtained using the Nikon AX R confocal imaging system.

### Retrospective extracardiac Fontan dataset

For calcification analysis, we used a dataset of late-term clinical CT scans of the cardiothoracic cavity from the Japanese TEVG clinical trial provided by Tokyo Women's Hospital (TWMU IRB-198). This long-term follow-up study for patients who had completed participation in the original clinical trial used an opt out consent. The Japanese clinical trial did not incorporate a control group; thus, we retrospectively searched the electronic medical record database at our institution (Nationwide Children's Hospital; IRB14-00035) for all patients with an extracardiac PTFE Fontan conduit (GORE-TEX® Stretch vascular graft) who underwent non-enhanced or contrast-enhanced CT scan. This retrospective study has a waiver for consent. We excluded studies that had visible intraluminal contrast within the Fontan conduit lumen, which could interfere with mural calcium assessment due to volume averaging. We used contrast studies with a lack of visible contrast within the Fontan

conduit due to the early/aortic phase of contrast enhancement since the Fontan circuit tends to fill late following an upper extremity intravenous injection of contrast. We also selected patients without a stent, pacemaker lead, or other device that caused significant artifact that could potentially interfere with attenuation measurement. Patient demographics were also obtained, and the study included both sexes. The study selection flow chart is included in (Supplementary Fig. 2).

### Ovine TEVG implantation and non-contrast computed tomography

Dorset cross or Cheviot sheep, both male and female, were implanted with PTFE grafts (GORE-TEX® stretch vascular grafts) ($n = 9$) or bone marrow cell-seeded TEVGs ($n = 11$) as inferior vena cava interposition grafts at 4–8 months of age[10]. Animals from both groups were imaged with non-contrast CT scans at 4–6 years post-graft implantation. Under general anesthesia, CT images were acquired at the level of the respective graft types with a slice thickness of 0.625 mm, at 450 mA, and 120 kVp using a commercially available clinical scanner (Discovery PET/CT 690, GE Healthcare). A subset of sheep (one TEVG and one PTFE animal) was also imaged after injection of intravenous contrast to differentiate between intravenous contrast and calcification. Subsequently, all analysis was performed blind to animal treatment status.

### Scaffold Phantom acquisition

To evaluate the baseline Hounsfield Units of the PTFE grafts, a pre-implantation sample was encased in 1% agar gel and imaged by non-contrast CT in an identical protocol to the ovine studies outlined above.

### Image processing

Images were anonymized and analyzed by two independent investigators blinded to all details, including age, sex, and graft type. All images were reconstructed to a 5 mm slice thickness to be consistent across all imaging studies. The investigator manually segmented the volume of interest (VOI) using a DICOM viewer (PMOD Technologies, Zurich, Switzerland). For human subjects, ROIs were manually drawn from the proximal anastomosis to the pulmonary artery to capture the entire graft. For ovine subjects, the VOI was designated as the vessel spanning from the diaphragm to the right atrium. The individual ROIs were interpolated and stacked to generate a 3D VOI for each patient, which was quantitatively assessed for calcium burden using relative calcium mass. Specifically, relative calcium mass was computed using the product of the mean density (HU) for voxels ≥130 HUs and the total volume of calcium[28]. A CT imaging threshold of ≥130 HUs was used to identify calcium within the scaffold[29]. For contrast-enhanced studies, if any voxels within the lumen exceeded the calcium threshold (>130 HU), the scan was excluded from the study due to the inability to separate vascular contrast from calcification. Independent investigator measurements were averaged and used for comparisons. Relative calcium mass was selected as the preferred analytical approach for standardizing the quantification of calcium burden in vascular scaffolds due to the retrospective nature of some clinical imaging studies included in the study as well as the use of multiple CT scanners at multiple institutions that spanned more than a decade.

### Histological analysis

Von Kossa staining was used to visualize inorganic phosphate molecules and calcium deposition in the scaffold and neotissue. Quantitative computerized image analyses were performed with ImageJ software (NIH, MD, USA) to detect positive staining in the samples. To avoid background signal, the analysis was manually restricted to the vessel wall area. % area positive stain was calculated by taking the total area of positive stain divided by the graft area (adventitia to lumen).

## Compliance and distensibility

Five-to-seven years post-implantation, the same sheep cohort underwent fluid overload stress testing. Two sheaths were inserted into the right internal jugular vein to guide the placement of two catheters: one into the abdominal IVC for contrast injection and one into the midgraft region for pressure measurement. 3D angiography and intra-graft pressure measurements were obtained concurrently before and after the administration of a saline bolus (20 ml/kg). All images were analyzed using Osirix MD® (Pixmeo SARL, Geneva, Switzerland) to determine cross-sectional area and circumference at five locations: low intrathoracic IVC (5 mm below the proximal anastomosis), proximal anastomosis, midgraft, distal anastomosis, and high intrathoracic IVC (5 mm above the distal anastomosis). The proximal and distal anastomoses were identified through surgically placed radiopaque markers. Compliance (2) and distensibility (3) were calculated using the following equations:

$$Compliance = \frac{A_{post} - A_{pre}}{P_{post} - P_{pre}} \left[ \text{cm}^2/\text{mmHg} \right] \qquad (2)$$

$$Distensibility = \frac{\frac{C_{post} - C_{pre}}{C_{pre}}}{P_{post} - P_{pre}} \times 10^4 [\%/100 \text{mmHg}] \qquad (3)$$

where $A$ is cross-sectional area (cm²), $C$ is circumference (cm), and $P$ is pressure (mmHg). Note that the pressure difference due to a bolus injection was $11.4 \pm 3.15$ mmHg for both classes of grafts, thus facilitating comparisons between TEVG and PTFE.

## Statistical analysis

Statistical analysis was performed in GraphPad Prism software (Version 9.2.0, GraphPad Software, CA, USA). Linear correlations were analyzed using Pearson's $R^2$ coefficient. Independent groups (i.e. TEVG vs PTFE implants) were compared using unpaired t-tests, with or without Welch's correction, or Mann Whitney U non-parametric test as appropriate pending normality and equivariant testing. A $p$-value of 0.05 was considered statistically significant.

## Computational modeling

We used computational hemodynamics to investigate how differences in compliance in the ovine IVC interposition grafts affected local blood pressure and flow. Noting that both the TEVG and PTFE grafts were subject to low, near-steady pressures under normal and bolus-injection conditions, a compliant graft necessarily distends more than a stiff graft when subjected to a comparable pressure. To simulate differences in pressure-induced distension, we prescribed the measured geometries in idealized 3D models of the host-graft-host segments under steady flow rates corresponding to normal and bolus flow.

We assumed that, under normal conditions, inferior vena cava (IVC) flow accounts for 70% of the cardiac output. The mean measured cardiac output was 80 ml/s across the PTFE and TEVG cohorts, yielding a calculated mean IVC flow rate of 56 ml/s. We applied this as an inlet Dirichlet boundary condition by mapping the flow rate to a parabolic velocity profile. We applied a resistance value at the outlet of 119 dynes · s/cm⁵ to yield a pressure of 5 mmHg at the outlet, a previously measured ovine atrial pressure. Additionally, we assumed that both the PTFE and TEVG grafts were optimally matched in diameter to the host IVC under normal conditions. We used an idealized diameter of 6.75 mm, representative of the average clinical geometry observed in the PTFE and TEVG cohorts under normal conditions. Under these assumptions, the PTFE and TEVG cohorts exhibited identical geometries under normal flow conditions.

Under bolus conditions, we modeled how differences in compliance between PTFE and TEVG grafts affected the distended geometry of the interpositional complex. Once again, we considered the

IVC flow to be 70% of the mean cardiac output, which under bolus conditions was 155 ml/s across the PTFE and TEVG cohort, resulting in a calculated IVC flow rate of 108.5 ml/s. We applied this as an inlet Dirichlet boundary condition by mapping the flow rate to a parabolic velocity profile. We maintained the resistance value at the outlet of 119 dynes · s/cm⁵ which yielded a 9.68 mmHg pressure at the outlet. Consistent with clinical observations, we modeled the TEVG as having distended similarly to the distal and proximal IVC vasculature under bolus conditions, reflecting the matching compliance that was observed to result in a 70% increase in cross-sectional area in both the IVC and midgraft in TEVG subjects. We thus increased the IVC and TEVG diameter by 30%, which was then fixed during the simulations. Conversely, we modeled the PTFE graft as unable to change geometry between normal and bolus conditions, consistent with the clinical observations that the midgraft had an approximately 0% cross-sectional area change under bolus conditions. Hence, we increased only the surrounding IVC cross-sectional area by 270% by increasing the diameter by approximately 92%. This again matched the average cross-sectional area change observed for the PTFE grafts under bolus conditions.

We then compared the hemodynamics of the TEVG and PTFE grafts under normal and bolus conditions. Simulations were completed using the open-source code SimVascular, release version May 2023, (https://simvascular.github.io/) to solve the 3D time-dependent incompressible Navier–Stokes equations on the non-moving, prescribed geometric domain where blood was modeled as an incompressible Newtonian fluid with a density of 1.06 g/cm³ and a viscosity of 0.04 dynes/cm². All models were discretized with 573,440 linear hexahedral elements and simulations were run with a timestep size of 0.01 s until a steady-state solution was achieved.

## Reporting summary

Further information on research design is available in the Nature Portfolio Reporting Summary linked to this article.

# Data availability

The data supporting the findings of this study are available within the article and its Supplementary Information files. The calcification, compliance, distensibility, and CFD data generated in this study are provided in the Supplementary Information and Source Data file. Source data are provided with this paper.

# Code availability

Simulations were completed using the open-source code SimVascular, release version May 2023, available at https://simvascular.github.io.

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

## Acknowledgements

Biospecimens were provided by the Pediatric Division of the Cooperative Human Tissue Network (CHTN), which is funded by the National Cancer Institute. Ex vivo data collection was facilitated by the Department of Pathology and Laboratory Medicine at Nationwide Children's Hospital, namely through contributions by Dr. Nilsa Ramirez and Miraj Patel. The authors extend their appreciation to the Histopathology Core at Nationwide Children's Hospital for their assistance in embedding, sectioning, and staining pathological samples. Further, the authors acknowledge the Nationwide Children's Hospital Animal Resource Core for their diligent and humane care of the animals involved in the study. This research was supported by the Department of Defense Award Number W81XWH-22-1-0597 in addition to NIH grants: R01 HL163065, R01 HL139796, and UH3 HL148693 awarded to CKB. WES was supported by The Ohio State University College of Medicine Roessler Research Scholarship.

## Author contributions

M.E.T., K.M.B., T.W., E.L.S., and M.N. co-wrote the manuscript and led experiments. J.T.L, D.J.V, W.E.S. and T.H.C. assisted with experiments. P.B.B., G.M., R.K., A.R.Y., K.N.H., J.D.H., A.L.M., M.R.S. and T.S. provided input in respective areas of the study and helped to design experiments. C.K.B. conceived of the study and supervised the performance of the study. All authors reviewed and revised the manuscript.

## Competing interests

Christopher Breuer and Toshiharu Shinoka received grant support from Gunze Limited, Pall Corporation, and Cook Regenetec. The remaining authors declare no competing interests.

## Additional information

[1]Center for Regenerative Medicine, Research Institute at Nationwide Children's Hospital, Columbus, OH, USA. [2]Molecular Cellular and Developmental Biology Graduate Program, The Ohio State University, Columbus, OH, USA. [3]The Ohio State University College of Medicine, Columbus, OH, USA. [4]Department of Biomedical Engineering, Yale University, New Haven, CT, USA. [5]Biomedical Sciences Graduate Program, The Ohio State University College of Medicine, Columbus, OH, USA. [6]Department of Exercise and Health Science, National Taipei University of Nursing and Health Sciences, Taipei City, Taiwan. [7]Pathology Department at Nationwide Children's Hospital and The Ohio State University, Columbus, OH, USA. [8]Department of Medical Safety Management, Tokyo Women's Medical University, Tokyo, Japan. [9]Department of Radiology, Nationwide Children's Hospital and The Ohio State University, Columbus, OH, USA. [10]Department of Pediatrics, The Ohio State University College of Medicine, Columbus, OH, USA. [11]The Heart Center, Nationwide Children's Hospital, Columbus, OH, USA. [12]Departments of Pediatrics and Bioengineering, Stanford University, Stanford, CA, USA. [13]Department of Surgery, The Ohio State University College of Medicine, Columbus, OH, USA. [14]Interdisciplinary Biophysics Graduate Program, The Ohio State University, Columbus, OH, USA. [15]Department of Cardiothoracic Surgery, Nationwide Children's Hospital, Columbus, OH, USA. [16]Department of Surgery, Nationwide Children's Hospital, Columbus, OH, USA. [17]These authors contributed equally: Mackenzie E. Turner, Kevin M. Blum, Tatsuya Watanabe, Erica L. Schwarz, Mahboubeh Nabavinia. ✉e-mail: Christopher.breuer@nationwidechildrens.org

