## [Peer Review File · Nature Communications]

REVIEWER COMMENTS

Reviewer #1 (Remarks to the Author):

This is an important study demonstrating the durability of vascular engineered grafts in comparison with PTFE grafts, particularly for what concerns the calcific degeneration. The study has a strong translational character with animal models and retrospective clinical evaluations. This is one of the limitations already well acknowledged by the authors. The article is well written and presented. I have several comments:

- 1.-The model may be only suitable for patients with specific congenital heart diseases and implanted in low pressure system. Other models have not been so successful in terms of durability when they are implanted in the left-side (J Vasc Surg. 2020 Jul;72(1):305-317.e6. doi: 10.1016/j.jvs.2019.06.221.). Would you have data in other congenital heart diseases where the pressure through the vascular graft would be higher?
- 2.-Is the patency of the vascular graft different according to the size of the graft (particularly smaller diameters)? do you have data on that to comment on?
- 3.-Table 1 misses the units for the time - is it years as in the other tables? From that table one can understand that time also influences the results and as you acknowledged in the study limitations section, the retrospective analysis precludes from assessing that influence. How were the patients selected retrospectively? were they all consecutive? Please indicate in a flow chart the exclusions and the reasons for those exclusions.
- 4.-Obviously, the retrospective analysis does not allow for correcting for reasons why the CT was performed. From Table 1 one can understand that there is an important bias in that sense. Is it true that specimen 5 who received a vascular graft at age 1.7 years did not have any repeated CT until 20 years later (if that is years because it is not clear)? could the authors provide data on sequential CTs at least for a subset of the patients who may have it?
- 5.-We do not know if the patients included in this analysis had any event between the implantation and the CT prior to degeneration of the device: were there endocarditis? were there interventions? please indicate whether the patients remained stable or not during the follow-up

Reviewer #2 (Remarks to the Author):

This manuscript provides important information on the hemodynamic performance, distensibility, and rate of calcium accumulation in the authors' tissue-engineered vascular graft (TEVG) implanted in sheep as an interposition graft in the systemic venous system with a comparison to comparably sized PTFE grafts. Accompanying this animal work are observations on calcium accumulation in analogous clinically implanted tissue engineered grafts and PTFE grafts in the venous system as part of a Fontan procedure using CT methods as well as tissue levels of calcium in PTFE conduits explanted from patients. The major important findings are that there was little calcium accumulation in the TEVG grafts in both the sheep and human TEVG samples, which was in contrast to evidence of significant calcium accumulation in the walls of the PTFE grafts. In the animal model, the TEVG grafts showed greater distensibility and lower pressure gradients in response to an intravenous fluid bolus compared to the PTFE grafts. This report builds on a series of prior reports from this group with this report focused on calcification and hemodynamic performance.

While the observation of markedly lower calcium deposition in the TEVG graft over time in vivo is an important one, the conceptual linkage to the difference between the hemodynamic performance of PTFE grafts and that of the TEVG grafts as related to calcification created some confusion in this reader's mind. It is important that the TEVG remain distensible in vivo, and this is likely related to the lack of accumulation of large amounts of calcium in these grafts. However, the non-distensibility of the PTFE grafts is inherent to the materials and fabrication of these grafts and is unrelated to the presence of the calcification in the wall, unless there is encroachment of the calcium into the lumen. This luminal encroachment is not commonly an issue with the PTFE grafts, and most narrowing is related to a build up of neo-intimal tissue on the luminal surface. From this perspective, the important message is the retained distensibility of the TEVG grafts over time compared to PTFE, and less the fact that non-distensible PTFE grafts also incite calcification. To the extent that this comparison of graft performance relies heavily on the calcification issue can result in a somewhat confusing message.

The calcification issue is clearly accentuated for other applications of tissue engineering to cardiovascular structures which undergo much larger deformations at much higher frequency, particularly heart valve leaflets, and valve leaflet calcification is almost always associated with valve dysfunction. The focus on the lack of calcification on the TEVG using these materials and the recellularization technique is important if these materials will be used in applications where large deformations and strains are necessary for normal function.

One minor issue with Figure 3 is that I found that the lines indicating a statistical comparison initially confused me, and this might be improved by placing a small downward extension of the line at each end indicating a comparison between two groups is being made.

As there is significant variability in the fabrication of PTFE grafts, it would also be useful to describe the source of these grafts for both the clinical and animal specimens. Changes in the fabrication methods have resulted in alterations of the luminal surface topography of these grafts, which have resulted in significant alterations in the host responses, particularly thrombogenicity and subsequent inflammation

Reviewer #3 (Remarks to the Author):

The authors investigate and compare the long-term formation of dystrophic calcification after PTFE and TEVG graft implantation in patients who received Fontan extracardiac operation. The study extends to validate clinical results with a long-term animal model and assesses the effects of calcification on the biomechanical properties. Authors also demonstrate the hemodynamic consequences of PTFE and TEVG implantations using computational fluid dynamics (CFD) models. The data presented in the manuscript is detailed and transparent. Their findings are significant, indicating a pronounced advantage of TEVGs over PTFE grafts for long-term outcomes. There are areas in the methodology that require additional detail to fully ensure the reproducibility of the results. It is recommended that the authors provide a more detailed explanation in the following sections for enhanced clarity.

1. In the final sentence of Introduction, the authors claim to assess the impact of dystrophic calcification and compliance mismatch on graft performance using computational fluid dynamics (CFD) simulations, referencing an idealized model of the conduit. However, the computational model does not appear to incorporate calcification or compliance factors; instead, it simply enlarges the conduit's size to represent post-bolus injection geometry. To avoid confusion, I recommend the authors refine this statement to accurately reflect the methodology employed.

2. In Figures 4B and C, there is notable variation in the compliance and distensibility of the TEVGs. Table 3 indicates that the age at implantation and the duration of implantation prior to CT imaging are very similar across the TEVG cohort, suggesting these factors do not contribute to the large standard deviation. Did the authors investigate if there is any correlation with the relative calcium mass? Such an analysis may provide deeper insight and strengthen the findings. Or it would be beneficial to explain other possible reasons for this large variation.

3. The velocity vector is presented in Figure 5 on a 2D plane. Could the authors clarify if this represents midplane velocities and whether a 2D or 3D models were used in the simulations?

4. In Results, Paragraph 5, how was the power loss calculated? It would be beneficial to include an equation.

5. In the Computational Modeling section, it is unclear whether the authors used an FSI model as they mention using "the open-source code svFSI". If they used an FSI model, further information on

the fluid-structure interaction settings should be provided. Moreover, while the PTFE graft is modeled as rigid, clarification on the material properties used to represent the IVC and TEVG would help in understanding the simulation framework. Did the authors perform time-dependent or steady simulations? Please provide further details on the time step and mesh size utilized in the simulations as well as details on what type of boundary conditions were specified at the inlet and outlet of the computational model.

Minor correction:

1. In Table 1, please provide the units for “Duration of Graft Implantation Prior to CT”.
2. In Results, Paragraph 5, dynes/cm²

REVIEWER COMMENTS

We would like to thank the reviewers for their thorough review of our manuscript.

Before responding to specific comments, we would like to acknowledge that during the process of preparing this revision we identified a discrepancy between the data in Table 3 and its graphical representation in Figure 3. Please find attached the revised graph, which now accurately portrays the ovine data.

We have also reviewed all our source materials related to this study to ensure that there are no other discrepancies. We apologize for this inadvertent error. Importantly, we want to emphasize that this correction does not alter any of the study's conclusions or interpretations.

Reviewer #1 (Remarks to the Author):

This is an important study demonstrating the durability of vascular engineered grafts in comparison with PTFE grafts, particularly for what concerns the calcific degeneration. The study has a strong translational character with animal models and retrospective clinical evaluations. This is one of the limitations already well acknowledged by the authors. The article is well written and presented. I have several comments:

1.-The model may be only suitable for patients with specific congenital heart diseases and implanted in low pressure system. Other models have not been so successful in terms of durability when they are implanted in the left-side (J Vasc Surg. 2020 Jul;72(1):305-317.e6. doi: 10.1016/j.jvs.2019.06.221.). Would you have data in other congenital heart diseases where the pressure through the vascular graft would be higher?

The vascular graft described in this proposal is not suitable for implantation in the arterial circulation because there is a prohibitively high incidence of aneurism formation and graft rupture, therefore, we have not evaluated the conduit in the arterial circulation in any long-term large animal studies or clinical trials. Interestingly when we evaluated the PLA (PCLA) version of the TEVG graft as either an intrathoracic IVC interposition graft in an ovine model or

as an extracardiac Fontan conduit clinically we did note calcification but to a much lesser extent than the PTFE graft (unpublished). Finally, we are currently evaluating a different TEVG as a pulmonary artery conduit (slightly higher pressure) and have similarly not observed calcification 1 year after implantation based on non-contrast CT in an ongoing study (unpublished).

2.-Is the patency of the vascular graft different according to the size of the graft (particularly smaller diameters)? do you have data on that to comment on?

The extracardiac Fontan conduit ranges in size from 12 mm-24 mm in diameter. Results of our previously published clinical studies did not identify a difference in patency based on graft size (Shin'oka, T. *et al.* Midterm clinical result of tissue-engineered vascular autografts seeded with autologous bone marrow cells. *The Journal of Thoracic and Cardiovascular Surgery* **129**, 1330–1338 (2005).)

3.-Table 1 misses the units for the time - is it years as in the other tables? From that table one can understand that time also influences the results and as you acknowledged in the study limitations section, the retrospective analysis precludes from assessing that influence. How were the patients selected retrospectively? were they all consecutive? Please indicate in a flow chart the exclusions and the reasons for those exclusions.

Thank you for identifying this oversight. This table has been corrected to include the units for time (years).

The studies were not consecutive. As requested, we provided a flow chart below. We will add the enclosed flow chart as a Supplement.

4.-Obviously, the retrospective analysis does not allow for correcting for reasons why the CT was performed. From Table 1 one can understand that there is an important bias in that sense. Is it true that specimen 5 who received a vascular graft at age 1.7 years did not have any repeated CT until 20 years later (if that is years because it is not clear)? could the authors provide data on sequential CTs at least for a subset of the patients who may have it?

We agree with the reviewer's comment that this is an important bias. The standard of care in assessment of Fontan patients is by echocardiogram followed by cardiac magnetic resonance imaging (CMR) when patients can undergo the study without sedation. The use of CT angiography (CTA) is not routine and the indications for CTA assessment are quite variable. In younger patients, a CTA study may be requested to assess the overall anatomy due to limitations of echocardiography and to avoid the need for anesthesia for CMR. In older patients, CMR is typically used for surveillance of the patient's ventricular volume, function, flow profiles and anatomy including the Fontan conduit, branch pulmonary arteries, aorta dimensions and collateral burden. Indications for CTA typically occurs when artifacts limit use of CMR, including metal artifacts from coils and stents, and typically to screen for thrombosis/embolism of the Fontan circuit, cause of cyanosis, assessment of branch pulmonary arteries typically difficult to visualized by echocardiography, Fontan baffle leak, systemic vein obstruction, pulmonary vein obstruction, aortic dilation, recurrent aortic obstruction, aortopulmonary collaterals, systemic-pulmonary vein collaterals, chylothorax, Fontan-associated lung disease and Fontan associated liver disease.

The use of CTA in Fontan patients is not routine due to concern of radiation exposure and CTAs in younger patients are used only when echocardiography is not diagnostic and cross-sectional

imaging is needed and unlike CMR, CTA can be performed with anesthesia in younger patients. In older patients, echocardiography remains the first line tool but the use of CMR is more routine when patients can undergo CMR evaluation without sedation. As such, the reasons for CTA assessment in Fontan patients varied significantly as detailed above. For our 6 PTFE subjects, a detail review for EMR for indications for CTA assessment is reported in the table below.

PTFE Subjects	Indication for CTA
CT PTFE1	Persistent pleural effusion
CT PTFE2	Abdomen mass/Unexplained pleural effusion
CT PTFE3	Assessment of LPA stent – not well visualized by echocardiography
CT PTFE4	Hemoptysis, concern for pulmonary embolism
CT PTFE5	Limited image quality by ECHO and Epicardial pacemaker contra-indicating the use of cardiac magnetic resonance imaging
CT PTFE6	Shortness of breath and orthopnea, assess for pulmonary embolism

Unfortunately, CTA is not routinely performed and when it does occur, the use of contrast is routine and opacification of the Fontan is typical resulting high luminal contrast. Since CTAs are not performed routinely, it is uncommon to have sequential CTA scans. We identified two PTFE subjects with two CTA scans each. Subject PTFE 3 had a CTA performed initially to assess the left pulmonary artery stent which was not well visualized by echocardiography and stent artifact limited use of CMR. The subject underwent balloon dilation of the LPA stent short after the CTA was performed due to stent narrowing. Due to timing of contrast in the Fontan lumen, a Relative Calcium Mass was not able to be performed. A follow-up CT was performed the day after the LPA stent (3 months since initial CTA) was balloon dilated due to concern of contrast extravasation. The Relative Calcium Mass was 932 HU*cm³ on the second study. Subject PTFE4 had two studies. The initial study was for hemoptysis and concern of pulmonary embolism. The follow-up study was for concern for restrictive lung disease. These studies were performed approximately 2 years apart with similar Calcium Mass Score of 680 HU*cm³ on the first study and 703 HU*cm³ on the second study.

5.-We do not know if the patients included in this analysis had any event between the implantation and the CT prior to degeneration of the device: were there endocarditis? were there interventions? please indicate whether the patients remained stable or not during the follow-up

We agree with the reviewer that processes including Fontan graft infection can contribute to alterations of the graft material, including degeneration. As noted above, we reviewed the EMR of the 6 PTFE Fontan subjects with indications provided in the table above. In review of the indication for CTA, the patients did not have issues with endocarditis. In addition, from the time of implantation to the CT and in two subjects with two CTAs, there was no report of endocarditis and the patient did not undergo an interventions for the Fontan. As noted above,

subject PTFE 3 did have dilation of the LPA stent between the two CTA studies with no intervention on the Fontan conduit. As such, no patients had any reported event or interventions related to the Fontan conduit in detail review of each subjects EMR.

Reviewer #2 (Remarks to the Author):

This manuscript provides important information on the hemodynamic performance, distensibility, and rate of calcium accumulation in the authors' tissue-engineered vascular graft (TEVG) implanted in sheep as an interposition graft in the systemic venous system with a comparison to comparably sized PTFE grafts. Accompanying this animal work are observations on calcium accumulation in analogous clinically implanted tissue engineered grafts and PTFE grafts in the venous system as part of a Fontan procedure using CT methods as well as tissue levels of calcium in PTFE conduits explanted from patients. The major important findings are that there was little calcium accumulation in the TEVG grafts in both the sheep and human TEVG samples, which was in contrast to evidence of significant calcium accumulation in the walls of the PTFE grafts. In the animal model, the TEVG grafts showed greater distensibility and lower pressure gradients in response to an intravenous fluid bolus compared to the PTFE grafts. This report builds on a series of prior reports from this group with this report focused on calcification and hemodynamic performance.

While the observation of markedly lower calcium deposition in the TEVG graft over time in vivo is an important one, the conceptual linkage to the difference between the hemodynamic performance of PTFE grafts and that of the TEVG grafts as related to calcification created some confusion in this reader's mind. It is important that the TEVG remain distensible in vivo, and this is likely related to the lack of accumulation of large amounts of calcium in these grafts. However, the non-distensibility of the PTFE grafts is inherent to the materials and fabrication of these grafts and is unrelated to the presence of the calcification in the wall, unless there is encroachment of the calcium into the lumen. This luminal encroachment is not commonly an issue with the PTFE grafts, and most narrowing is related to a build-up of neo-intimal tissue on the luminal surface. From this perspective, the important message is the retained distensibility of the TEVG grafts over time compared to PTFE, and less the fact that non-distensible PTFE grafts also incite calcification. To the extent that this comparison of graft performance relies heavily on the calcification issue can result in a somewhat confusing message.

We apologize for the confusion. You make an excellent point, the lack of distensibility of the PTFE graft is multifactorial and while it might be reduced by calcification, within the physiological range (0-20 mmHg), it is primarily driven by the material properties of the PTFE. We have revised the manuscript in an attempt to avoid this confusion.

The calcification issue is clearly accentuated for other applications of tissue engineering to cardiovascular structures which undergo much larger deformations at much higher frequency, particularly heart valve leaflets, and valve leaflet calcification is almost always associated with valve dysfunction. The focus on the lack of calcification on the TEVG using these materials and

the recellularization technique is important if these materials will be used in applications where large deformations and strains are necessary for normal function.

We agree with your comment and hope that this manuscript will spur further investigation in this important area investigation.

One minor issue with Figure 3 is that I found that the lines indicating a statistical comparison initially confused me, and this might be improved by placing a small downward extension of the line at each end indicating a comparison between two groups is being made.

We apologize for the confusion. Thank you for your suggestion, we have revised the figure accordingly.

As there is significant variability in the fabrication of PTFE grafts, it would also be useful to describe the source of these grafts for both the clinical and animal specimens. Changes in the fabrication methods have resulted in alterations of the luminal surface topography of these grafts, which have resulted in significant alterations in the host responses, particularly thrombogenicity and subsequent inflammation.

The PTFE grafts used in the study were GORE-TEX® stretch vascular grafts, now noted in the methods section.

Reviewer #3 (Remarks to the Author):

The authors investigate and compare the long-term formation of dystrophic calcification after PTFE and TEVG graft implantation in patients who received Fontan extracardiac operation. The study extends to validate clinical results with a long-term animal model and assesses the effects of calcification on the biomechanical properties. Authors also demonstrate the hemodynamic consequences of PTFE and TEVG implantations using computational fluid dynamics (CFD) models. The data presented in the manuscript is detailed and transparent. Their findings are significant, indicating a pronounced advantage of TEVGs over PTFE grafts for long-term outcomes. There are areas in the methodology that require additional detail to fully ensure the reproducibility of the results. It is recommended that the authors provide a more detailed explanation in the following sections for enhanced clarity.

Please see below.

1. In the final sentence of Introduction, the authors claim to assess the impact of dystrophic calcification and compliance mismatch on graft performance using computational fluid dynamics (CFD) simulations, referencing an idealized model of the conduit. However, the computational model does not appear to incorporate calcification or compliance factors; instead, it simply enlarges the conduit's size to represent post-bolus injection geometry. To avoid confusion, I recommend the authors refine this statement to accurately reflect the methodology employed.

To better clarify what was studied and measured, we changed the wording of the noted sentence to more specifically reflect that the simulations estimated the hemodynamic outcomes that arise primarily from the geometries of the TEVG and PTFE grafts under different flow conditions, noting that neither graft distends much in vivo due to the near steady flow, moderate-to-low pressures. The sentence now reads as follows:

“Finally, we evaluate the impact of compliance mismatch on graft hemodynamics. Despite similar geometries under normal flow conditions, the difference in compliance between the TEVG and PTFE grafts resulted in significantly different geometries during imposed bolus flow conditions. We use computational fluid dynamics to discover differential hemodynamic performance based on geometrically different 3D computational models of TEVG and PTFE graft-host IVC segments under steady flow rates corresponding to normal and bolus flow.”

This better emphasizes that the effects that calcification and compliance have on graft hemodynamics are mediated through the geometries produced under the aforementioned flow conditions, and that it is measured geometry which is being simulated.

2. In Figures 4B and C, there is notable variation in the compliance and distensibility of the TEVGs. Table 3 indicates that the age at implantation and the duration of implantation prior to CT imaging are very similar across the TEVG cohort, suggesting these factors do not contribute to the large standard deviation. Did the authors investigate if there is any correlation with the relative calcium mass? Such an analysis may provide deeper insight and strengthen the findings. Or it would be beneficial to explain other possible reasons for this large variation.

Based on your suggestion we performed the suggested correlations which we will include in a Supplement. In addition we have revised the manuscript to include possible reasons for the large variation in the compliance and distensibility.

3. The velocity vector is presented in Figure 5 on a 2D plane. Could the authors clarify if this represents midplane velocities and whether a 2D or 3D models were used in the simulations?

Yes, the figure caption has been updated to specify the midplane velocity field:

“Images showing (A.) the midplane flow-related velocity vectors of a TEVG versus a PTFE graft before and after a bolus injection with 20 ml/kg BW isotonic saline.”

In addition, the adjective “3D” has been added where appropriate in the manuscript to specify that the models used were 3D.

4. In Results, Paragraph 5, how was the power loss calculated? It would be beneficial to include an equation.

The power loss was calculated using the following equation which was also added and cited in the paper as Equation 3:

“The power loss was calculated as

$$\text{Power loss} = 1 - \frac{\int_{A_{out}} \left(p + \frac{1}{2} \rho u^2 \right) \mathbf{u} \cdot d\mathbf{A}}{\int_{A_{in}} \left(p + \frac{1}{2} \rho u^2 \right) \mathbf{u} \cdot d\mathbf{A}} \quad (3)$$

where A_{in} is the inlet the control surface, A_{out} is the outlet control surface, u is the magnitude of the velocity, \mathbf{u} is the velocity in the normal direction to the control surface, p is the static pressure, and ρ is the fluid density.”

5. In the Computational Modeling section, it is unclear whether the authors used an FSI model as they mention using “the open-source code svFSI”. If they used an FSI model, further information on the fluid-structure interaction settings should be provided. Moreover, while the PTFE graft is modeled as rigid, clarification on the material properties used to represent the IVC and TEVG would help in understanding the simulation framework. Did the authors perform time-dependent or steady simulations? Please provide further details on the time step and mesh size utilized in the simulations as well as details on what type of boundary conditions were specified at the inlet and outlet of the computational model.

Thank you for this comment. To clarify, the simulations were run on non-moving domains with wall geometry prescribed as the experimentally observed diameters of the graft complexes under the noted flow conditions. Although these simulations were solved as a non-moving domain, it may be unsuitable to label them as “rigid” simulations as the geometries were dynamically adjusted under specific flow conditions in order to match the observed geometric changes under those conditions. Calling them “rigid” may imply that their geometries do not change in response to experienced flow conditions, which is not true except for the case of the

PTFE graft where the term “rigid” refers to the known material properties that gave rise to the observed geometry (no change diameter under bolus conditions) that was then applied to the corresponding model. Therefore, rather than referring to these simulations as “rigid”, we instead refer to them as “prescribed geometry” simulations and have changed the vocabulary throughout the section to reflect this.

We also simulated steady-flow conditions. To better specify this, the first paragraph of the section was modified to read:

“We used computational hemodynamics to investigate how differences in compliance in the ovine IVC interposition grafts affected local blood pressure and flow. Noting that both the TEVG and PTFE grafts were subject to low, near steady pressures under normal and bolus-injection conditions, a compliant graft necessarily distends more than a stiff graft when subjected to a comparable pressure. To simulate differences in pressure-induced distension, we prescribed the measured geometries in idealized 3D models of the host-graft-host segments under steady flow rates corresponding to normal and bolus flow.”

We did use the svFSI code to solve for the hemodynamics in the prescribed geometry domain as the svFSI code contains a solver for a steady-domain as well. To prevent confusion, we cite the parent software suite “SimVascular” rather than “svFSI”.

To better specify the boundary conditions, we added the following text to the section:

“We applied this as an inlet Dirichlet boundary condition by mapping the flow rate to a parabolic velocity profile. We applied a resistance value at the outlet of $119 \text{ dynes} \cdot \text{s}/\text{cm}^5$ to yield a pressure of 5 mmHg at the outlet, a previously measured ovine atrial pressure.”

“We applied this as an inlet Dirichlet boundary condition by mapping the flow rate to a parabolic velocity profile. We maintained the resistance value at the outlet of $119 \text{ dynes} \cdot \text{s}/\text{cm}^5$ which yielded a 9.68 mmHg pressure at the outlet.”

Finally, we comment on the discretization resolution and the time-step size:

“All models were discretized with 573,440 linear hexahedral elements and simulations were run with a timestep size of 0.01 seconds until a steady-state solution was achieved.”

Minor correction:

1. In Table 1, please provide the units for “Duration of Graft Implantation Prior to CT”.
2. In Results, Paragraph 5, dynes/cm^2

Thank you for identifying these oversights. We have corrected them in the revised manuscript.

REVIEWERS' COMMENTS

Reviewer #1 (Remarks to the Author):

The authors have been very responsive and have amended the manuscript for clarification. Still, the limitations highlighted in the rebuttal have not been included in the study limitations section: the selection bias, the lack of data in models that include grafts in the arterial side,...the limitations that reviewer 1 has indicated need to be clearly included.

Reviewer #2 (Remarks to the Author):

This revised manuscript has addressed this reviewer's comments for the most part, but I would have preferred more attention to the distinction between calcification in vascular grafts and vascular graft "failure" unless the calcification process is associated with impingement on the lumen.

Reviewer #3 (Remarks to the Author):

Authors addressed comments effectively. Recommend to publish.

RESPONSE TO REVIEWERS

We appreciate the reviewers' careful review of our manuscript.

Reviewer 1

The authors have been very responsive and have amended the manuscript for clarification. Still, the limitations highlighted in the rebuttal have not been included in the study limitations section: the selection bias, the lack of data in models that include grafts in the arterial side,...the limitations that reviewer 1 has indicated need to be clearly included.

Thank you so much for your feedback.

The following has been added to the discussion of the manuscript, page 11 line 345:

"Importantly, this study focuses on Fontan conduits, a low pressure high flow system. These findings may not be directly applicable to high pressure systems such as the arterial circulation."

And page 12 starting at line 351:

"We were limited by the samples available to us, which raises the possibility of selection bias. However, we did not exclude any available samples on the basis of age, sex, race, etc."

Reviewer 2:

This revised manuscript has addressed this reviewer's comments for the most part, but I would have preferred more attention to the distinction between calcification in vascular grafts and vascular graft "failure" unless the calcification process is associated with impingement on the lumen.

We appreciate the reviewer's comments. The following has been added to page 10, Line 299 of the manuscript:

"Calcification is a relatively understudied complication associated with Fontan conduits and the progression to elevated conduit pressure. Calcification can also progress and cause stenosis, but this is a less frequent occurrence.³ Although none of the grafts in this study demonstrated clinical failure, calcification was present and our findings suggest that decreased compliance is detrimental to long-term performance."

We revised the introduction to include the following, beginning on page 2, line 61:

"Furthermore, dystrophic calcification can progress and impinge on the lumen, contributing to the formation of critical stenosis that requires treatment with angioplasty, stenting, or surgical graft replacement.³ Calcification may also complicate any necessary re-operation through calcific attachment to surrounding anatomy."